# Artificial 2D van der Waals Synapse Devices via Interfacial Engineering for Neuromorphic Systems

**DOI:** 10.3390/nano10010088

**Published:** 2020-01-02

**Authors:** Woojin Park, Hye Yeon Jang, Jae Hyeon Nam, Jung-Dae Kwon, Byungjin Cho, Yonghun Kim

**Affiliations:** 1Department of Advanced Material Engineering, Chungbuk National University, Chungdae-ro 1, Seowon-Gu, Cheongju, Chungbuk 28644, Korea; wjpark@chungbuk.ac.kr (W.P.); hyjang0581@gmail.com (H.Y.J.); jhnam0714@gmail.com (J.H.N.); 2Materials Center for Energy Convergence, Surface Technology Division, Korea Institute of Materials Science (KIMS), 797 Changwondaero, Sungsan-gu, Changwon, Gyeongnam 51508, Korea; jdkwon@kims.re.kr

**Keywords:** 2D heterostructure, WSe_2_, NbSe_2_, Nb_2_O_5_ interlayer, synapse device, neuromorphic system

## Abstract

Despite extensive investigations of a wide variety of artificial synapse devices aimed at realizing a neuromorphic hardware system, the identification of a physical parameter that modulates synaptic plasticity is still required. In this context, a novel two-dimensional architecture consisting of a NbSe_2_/WSe_2_/Nb_2_O_5_ heterostructure placed on an SiO_2_/p+ Si substrate was designed to overcome the limitations of the conventional silicon-based complementary metal-oxide semiconductor technology. NbSe_2_, WSe_2_, and Nb_2_O_5_ were used as the metal electrode, active channel, and conductance-modulating layer, respectively. Interestingly, it was found that the post-synaptic current was successfully modulated by the thickness of the interlayer Nb_2_O_5_, with a thicker interlayer inducing a higher synapse spike current and a stronger interaction in the sequential pulse mode. Introduction of the Nb_2_O_5_ interlayer can facilitate the realization of reliable and controllable synaptic devices for brain-inspired integrated neuromorphic systems.

## 1. Introduction

Continuous downscaling has stimulated the development of semiconductor technology for the last several decades, offering advantages, such as lower power consumption, higher integration, faster circuit operation, and reduced device cost per function. However, the side effects from continuous downscaling, to a size of less than 10 nm, limit the further development of the silicon semiconductor technology. This has motivated the exploration of novel computation systems beyond the conventional Von Neumann architecture that can overcome the downscaling limitations. Recently, due to the increasing need to implement sophisticated information processing system mimicking the human brain, the neuromorphic computing system has attracted a great deal of attention [1,2,3,4,5]. For the integrated neuromorphic systems, it is important to realize operations of complex and diverse functions implemented using a parallel architecture consisting of ~10^11^ neurons and ~10^15^ synapses. Additionally, the unit event should be simultaneously conducted using an extremely small amount of energy [6].

The artificial synapse device is considered to be an essential fundamental element for the emulation of biological neural networks [7]. The mechanism of operation for transmitting a spike input stimulus through the synapse can strengthen or weaken the synaptic weight, which is known as synaptic plasticity [8]. The synapse provides the functions of information processing and storage based on the spiking neural network. For this system, conventional solid-state electronics technology has been adopted for emulating the biological synapse function, in order to demonstrate a neuromorphic computing system [9]. In previous studies, conventional silicon-based complementary metal-oxide semiconductor (CMOS) technology was employed for demonstrating solid-state synapse devices, and a network consisting of 256 million configurable synapses and 1 million programmable spiking neurons was demonstrated [10]. The use of the 28-nm fully depleted silicon-on-insulator CMOS technology for 64k-synapse and 256-neuron architecture was also reported [11]. However, these CMOS-based devices are still unsuitable for realizing an artificial intelligence chip, because they cannot meet the requirements of higher integration density and lower power consumption. Si CMOS-based synapse device is based on the operation of complex logic circuits. This means that its power dissipation is essentially higher than that of other types, which is not satisfactory for emulating the biological synapse with an ultralow femtojoule energy consumption.

To eliminate the bottlenecks hindering the further development of neuromorphic computing systems, three-terminal artificial synaptic transistors, based on novel semiconductors have been studied to demonstrate synaptic functions. For instance, diverse semiconducting materials including carbon nanotubes, [12] nickelate, [13], and indium gallium zinc oxide (IGZO) [14,15] have been selected for the realization of synapse platforms. Meanwhile, two-dimensional (2D) transition dichalcogenides (TMDCs) are an intriguing nanomaterial layer for key elements of synaptic transistors due to their advantages of excellent intrinsic scalability, transparency, chemical robustness, and low power consumption [16,17,18,19]. In fact, several research groups have demonstrated the corresponding synaptic devices [20,21]. Meanwhile, a variety of oxide layers have been used as the conductance-tuning layers for synapse device applications. For example, phase change memory emulating synaptic behavior was demonstrated using a thin HfO_2_ interface layer [22]. Additionally, Deswal et al. reported an NbO_x_-based memristor, showing a gradual and continuous conductance change that is a prerequisite of a biological synapse device [23]. Nevertheless, it is still unclear what physical parameters can be used to precisely manipulate the synaptic functions. Thus, the use of a 2D heterostructure, combined with insulating oxide, can be an alternative approach for the development of energy-efficient artificial synapse devices.

In this work, we designed a vertically-stacked 2D metallic electrode NbSe_2_/semiconductor WSe_2_/interlayer Nb_2_O_5_ heterostructure placed on an Si/SiO_2_ substrate with the back-gate configuration. Here, WSe_2_ and Nb_2_O_5_ served as the active channel, and the conductance-tuning layer, respectively. Additionally, the NbSe_2_ electrode can provide excellent transistor switching characteristics due to a sharp 2D interface and the absence of the metal-induced gap states [24,25]. The post-synaptic current behavior can be modulated precisely by adjusting the thickness of the Nb_2_O_5_ layer, with a thicker Nb_2_O_5_ interlayer providing higher synapse spike current and strong interaction in paired pulse facilitation testing modes. The charge trapping/detrapping mechanism at the Nb_2_O_5_ defect states based on an energy band model was proposed. The novel 2D architecture will pave the way toward extreme integration for the development of the massively parallel neuromorphic circuitry system.

## 2. Materials and Methods

### 2.1. CVD Synthesis of WSe_2_ and NbSe_2_

A selenium (Se)-based semiconducting channel based on WSe_2_ and a metallic electrode based on NbSe_2_ were synthesized using a simple two-step process. First, WO_3_ and Nb_2_O_5_ thin films were individually deposited on an SiO_2_/Si wafer. The thicknesses of the WO_3_ and Nb_2_O_5_ thin films were ~3, and ~5 nm, respectively. This pre-deposited oxide layer on the wafer was directly loaded into the center of thermal furnace and vacuumed with a rotary pump system. Then, the thermal furnace was heated to the desired temperature (~1000 °C) under the flow of 5% hydrogen-balanced Ar gas (Ar/H_2_), while a selenium powder source was sublimated by heating to 500 °C. After a 1-h selenization process, the furnace was naturally cooled down to room temperature.

### 2.2. Fabrication of 3-Terminal Synapse Device

A heavily doped p-type Si substrate with SiO_2_ was cleaned by sonication in acetone, methanol, and iso-propyl alcohol (IPA) solution. To precisely tune the synaptic weight corresponding to the drain current, the charge trapping layer of the Nb_2_O_5_ thin film was deposited with different thicknesses using thermal evaporation. The thickness of Nb_2_O_5_ varied from 2.6 to 3.9 nm, as validated by the cross-sectional transmission electron microscopy (TEM) analysis. Then, the synthesized WSe_2_ semiconducting channel was transferred onto an SiO_2_/Si wafer using a poly(methyl methacrylate)-assisted transfer method and patterned using conventional photolithography. Finally, the NbSe_2_ metallic electrode was transferred for the formation of the NbSe_2_/WSe_2_ van der Waals heterojunction, in order to minimize the contact resistance [24,25].

### 2.3. Electrical Characterization

Basic electrical characterizations were carried out using a Keithley 2636B source meter (Keithley Instruments, Solon, OH, USA). The amplitude of the applied synaptic pulse, used to generate an excitatory post-synaptic current (EPSC), was 20 V and its duration was varied from 2 to 10 s.

## 3. Results and Discussion

Figure 1a shows a schematic of a biological neural network consisting of synapses and neurons. The most important trait of brain-inspired devices is their capability for efficient data processing using an extremely small amount of power in the networks with an astronomical number of synapses and neurons. The parallel network means that processing and storage of information occur simultaneously and do not follow the von Neumann computing paradigm. Therefore, a high device integration density and low energy consumption are crucial for a neuromorphic system. The operation of transmitting a spike input stimulus is illustrated in Figure 1b. The interaction of the pre- and post-synaptic activities affects the long-lasting connection strength, and long-lasting plasticity is considered to be the key mechanism of basic neuromorphic computation. Figure 1c shows the back-gate configuration of the WSe_2_ synapse transistor. The heavily-doped Si layer was used as the back-gate and NbSe_2_ was used as the source/drain. The Nb_2_O_5_ interfacial layer allows the fine-tuning of the conductance of the WSe_2_ transistor.

Figure 2a shows a schematic of the electrical measurements of the synapse device in the back-gate pulse system. Figure 2b shows the obtained cross-sectional high-resolution transmission electron microscopy images and the results of the energy-dispersive X-ray spectroscopy (EDS) analysis, thereby, clearly demonstrating the distinct film layers and sharp junction interfaces. The different stacking structures of WSe_2_-NbSe_2_, 2.6 nm Nb_2_O_5_-WSe_2_-NbSe_2_, and 3.9 nm Nb_2_O_5_-WSe_2_-NbSe_2_ were clearly observed and compared. The boundaries of each layer appeared to be atomically sharp and smooth without a significant interfacial gap. Five layers of NbSe_2_ and three layers of WSe_2_ were consistently observed for all of the samples, and the additional interfacial Nb_2_O_5_ layer was also clearly observed. The distributions of the W, Se, Nb, and O elements were obtained from the EDS elemental mapping images. The left panel of Figure 2b shows the WSe_2_-NbSe_2_ stack architecture without the Nb_2_O_5_ deposition. Since, both the NbSe_2_ and Nb_2_O_5_ films contain Nb atoms, the two separate Nb layers were observed only in the samples with the Nb_2_O_5_ interfacial layer, verifying the existence of Nb_2_O_5_. The middle panel of Figure 2b shows the results for the sample with a 2.6 nm Nb_2_O_5_ layer. The right panel of Figure 2b shows the sample with a 3.6 nm Nb_2_O_5_ layer. Figure 2c shows that the Raman spectra obtained for the as-synthesized 2D films support the presence of 2D materials, such as WSe_2_ and NbSe_2_, demonstrating the successful synthesis of the 2D nanomaterials via the chemical vapor deposition (CVD) technique. The Raman spectra of WSe_2_ and NbSe_2_ clearly display the in-plane vibrational modes of W-Se and Nb-Se (E^1^_2g_: 250.3 and 243.2 cm^−1^) and the out-of-plane vibrational modes that arise from the motion the Se atoms (A_1g_: 258.5 and 230.6 cm^−1^ for WSe_2_ and NbSe_2_). Furthermore, two distinct Raman peaks of WSe_2_ and NbSe_2_ with stacked device structure were also observed even after transfer process in Appendix A.

To compare the transfer characteristics of the WSe_2_-NbSe_2_ van der Waals hetero-junction devices with different Nb_2_O_5_ thickness, DC-mode-based double sweep measurements were performed, as shown in Figure 3a. The double sweep curves of the 2D heterojunction devices were obtained under varying values of able V_BG_ in the range from 10 to −20 V at a fixed drain voltage of −5 V. The WSe_2_-based transistor showed typical p-type unipolar behavior, with a counterclockwise hysteresis loop, that may be ascribed to the confinement of the hole charges in the trap states induced by the Nb_2_O_5_ interlayer [26]. Additionally, the repeatability test of DC transfer double sweep curves, with different Nb_2_O_5_ thicknesses, were also shown in Appendix A. We also investigated the statistical distribution of the hysteresis window voltages, in order to validate the reliability of the data corresponding to the hysteresis behavior (Figure 3b). The average values of the hysteresis voltage for each device were measured to be ~5, 7, and 11 V, respectively. The value of the error bar was almost same for all of the devices. Thus, it is clear that a thicker Nb_2_O_5_ interlayer gives rise to a larger hysteresis window. The dependence of DC sweep speed on transfer curves was also depicted in Appendix A.

To elucidate the origin of the hysteresis of the 2D heterostructure transistors, the corresponding energy band model was proposed (Figure 4). We previously reported the positive effect of the combination of WSe_2_-NbSe_2_ with reduced contact barrier [24,25]. The conventional Richardson-Schottky equation was employed to calculate Schottky barrier,
(1)$$I_{DS}\;=\;AA^{*}T^{2}\exp\left[-\frac{\left(\Phi_{B}-\sqrt{q^{3}V/4\pi \varepsilon_{o}\varepsilon_{r}d}\right)}{k_{b}T}\right]$$
where *A* is the contact area, *A** is the effective Richardson constant, *T* is the temperature, Φ*_B_* is the Schottky barrier height, *q* is the electron charge, *V* is the applied forward bias, *ε*_0_ and *ε_r_* are the permittivity of the vacuum and the oxide layer, respectively, *d* is the width of the interface barrier, and *K_b_* is the Boltzmann constant. It was mentioned in the references that Schottky barrier at WSe_2_-NbSe_2_ contact is significantly lower than that at WSe_2_-metal(Pd) contact due to Fermi-level de-pinning. Therefore, the 2D WSe_2_-NbSe_2_ combination can be an excellent candidate for the fabrication of an energy-efficient low-power synaptic transistor, due to its low contact resistance. Recently, the new methodology for universal 2D material was reported to obtain Schottky barrier, suggesting more accurate calculation [27]. Holes are known to be the major carriers in both the semiconductor channel WSe_2_ and the metallic source/drain electrode NbSe_2_. Thus, only the hole charge transport was considered in our proposed switching model. As shown in Figure 4a, the negative voltage applied to the back gate electrode (p +Si) shifts the corresponding Fermi level upward, accumulating hole charge near the Nb_2_O_5_-corresponding defect states. Under a negative gate bias, holes can be easily trapped in the defect states within the Nb_2_O_5_ interlayer, depleting the carriers in the WSe_2_ and leading to a decrease in the drain current. Meanwhile, when a positive voltage is applied to the gate, the Fermi level shifts downward, depleting the trapped holes in the Nb_2_O_5_ defects (Figure 4b). Simply put, the trapped holes will be released across the Nb_2_O_5_-WSe_2_ interface, leading to an increase in the drain current. Indeed, we experimentally proved that the amount of the trapped hole carriers is controlled by the Nb_2_O_5_ thickness.

To characterize the pulse response of the 2D heterostructure devices, we monitored the spike current response to the gate voltage pulses with the amplitude and duration time of 20 V, and 2 s, respectively (Figure 5a). In neuroscience, it is important to transfer electrical or chemical signal from pre-synapse to post-synapse. This is usually caused by the flow of positively charged ions. EPSC can be generated by the action of ions or electron flow in the neuromorphic system. The gate voltage for the EPSC was fixed at −20 V to give a fair comparison for each case. EPSC reaches the maximum value and then decays back to the initial current state. Interestingly, the spike was generated, even in the reference device without Nb_2_O_5_ layer. This might be because of the unintentional charge trap sites, which exist at the diverse interfaces (WSe_2_-NbSe_2_ and SiO_2_-WSe_2_). Our result showed that the peak values increase with the increasing thickness of the inserted Nb_2_O_5_ interlayer. Higher voltage pulses required long decay time to restore the synapse device to the initial current state, leading to stronger nonvolatile properties. The duration time of the pulse voltage, that is applied to the devices also affected the peak EPSC (Figure 5b). A longer pulse duration resulted in a higher peak EPSC. In a biological neural network, paired pulse facilitation is an important synapse parameter for determining synaptic plasticity, that is responsible for learning and memory processes [28]. As shown in Figure 5c, paired pulse facilitation is the phenomenon where the EPSC stimulated by the second spike is enhanced when the first spike is closely followed by the second spike [29,30]. Such essential synapse behavior can be emulated using our 2D heterostructure transistor. Figure 5d shows the interaction of two sequential spikes for all of the devices. The interval time between the applied pulses was 2 s. The interaction between the output spike current of the control device without Nb_2_O_5_ was not observed, indicating the negligible synaptic weight modulation property. Meanwhile, the introduction of the Nb_2_O_5_ layer strengthened the interaction of the two spikes; more specifically, a thicker Nb_2_O_5_ interlayer induced a much greater current change at the second pulse mode. Such a strong tuning ability of the synaptic weight enhances the electrical plasticity of the artificial synapse device, and may improve the intelligence of the integrated neuromorphic system [12].

## 4. Conclusions

We demonstrated controllable synaptic plasticity with the WSe_2_/Nb_2_O_5_ heterostructure in the WSe_2_ back-gate device. The Nb_2_O_5_ layer served as the conductance-modifying layer and enabled precise modulation of the conductive states and their dynamic change. Essential synaptic functions (EPSC and paired pulse facilitation) were investigated in the WSe_2_/Nb_2_O_5_ heterostructure devices. In particular, the optimized thickness of the Nb_2_O_5_ layer strengthened the interaction in the synaptic weight, showing the largest post-synapse current. Thus, the facile one-step Nb_2_O_5_ layer deposition process, demonstrated in this work, is an effective approach for the realization of controllable synaptic devices.

## Figures and Tables

**Figure 1 nanomaterials-10-00088-f001:**
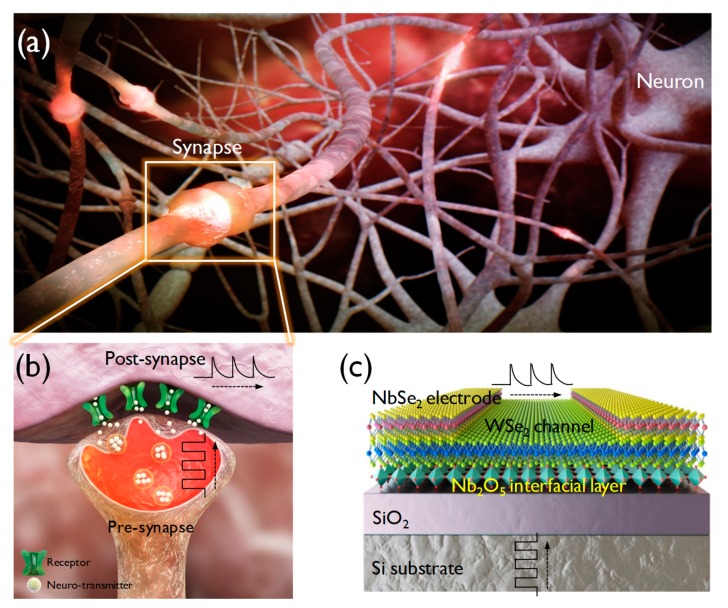
(**a**) Biological neural network consisting of synapses and neurons. (**b**) Operational mechanism of the transmission of an input stimulus from pre-synapse to post-synapse. (**c**) Artificial synapse transistor comprised by vertically stacked NbSe_2_/WSe_2_/Nb_2_O_5_/SiO_2_/p+ Si, mimicking the function of bio synapse.

**Figure 2 nanomaterials-10-00088-f002:**
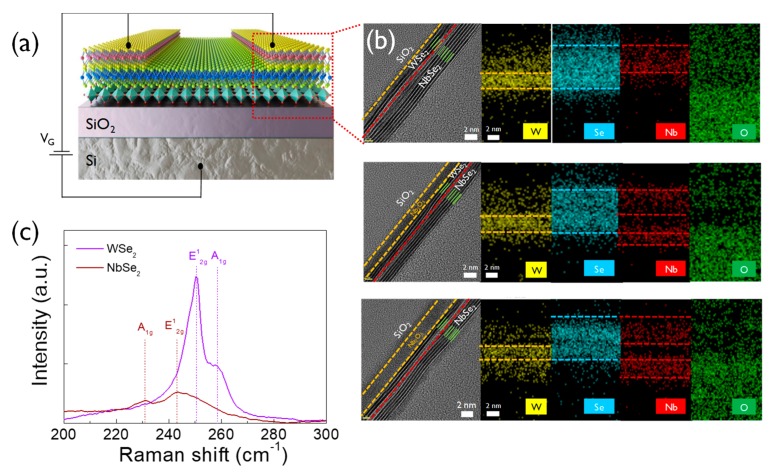
(**a**) Configuration scheme for the electrical measurements of the synapse transistor device. (**b**) Cross-sectional high-resolution transmission electron microscopy and energy-dispersive X-ray spectroscopy (EDS) elemental mapping images recorded from WSe_2_-NbSe_2_, 2.6 nm Nb_2_O_5_-WSe_2_-NbSe_2_ and, 3.9 nm Nb_2_O_5_-WSe_2_-NbSe_2_ (**c**) Raman spectra for WSe_2_, and NbSe_2_ that serve as the active channel and metallic electrode, respectively.

**Figure 3 nanomaterials-10-00088-f003:**
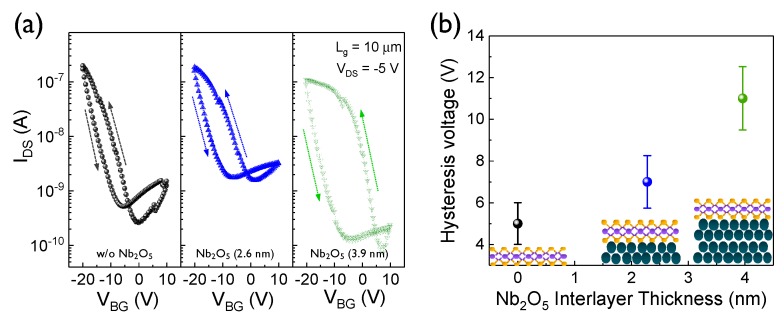
(**a**) Hysteresis behaviors of the two-dimensional (2D) WSe_2_-NbSe_2_ hetero-structure transistor devices with different Nb_2_O_5_ interlayer thickness. (**b**) Hysteresis window voltage as a function of the Nb_2_O_5_ interlayer thickness for the 2D WSe_2_-NbSe_2_ devices.

**Figure 4 nanomaterials-10-00088-f004:**
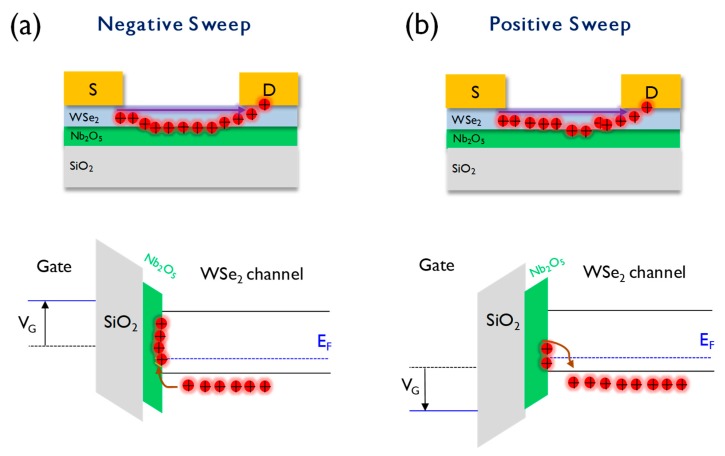
(**a**) Device operation scheme and energy band model of the 2D heterostructure transistor device for describing the trapping behavior of the hole carriers at the negative gate bias condition; (**b**) device operation scheme and energy band model of the 2D heterostructure transistor device, corresponding to the process of the release of the trapped hole carriers at a positive gate bias condition.

**Figure 5 nanomaterials-10-00088-f005:**
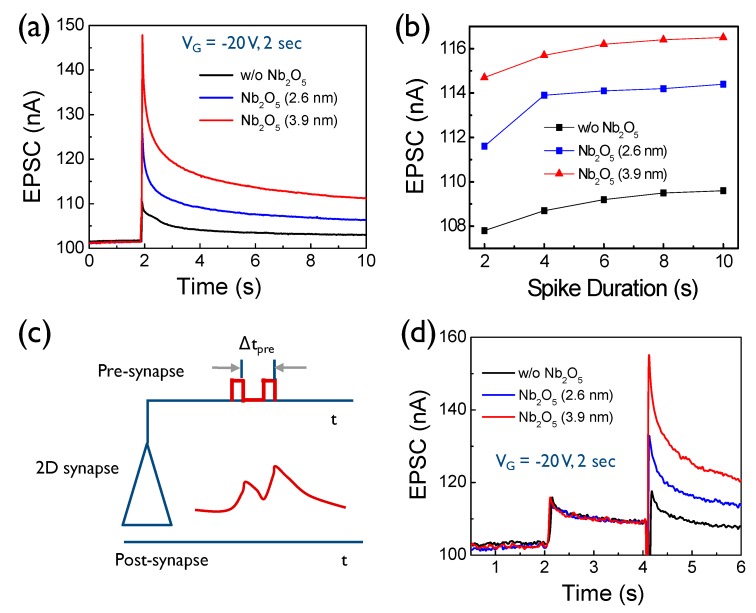
(**a**) Comparison of the excitatory post-synaptic current (EPSC) behavior of the 2D heterostructure devices with different Nb_2_O_5_ interlayer thicknesses. (**b**) Comparison of the EPSC data as a function of the spike duration time for the different 2D heterostructure devices. (**c**) Operation scheme of the synapse circuit for describing paired pulse facilitation that is stimulated by the application of two sequential pulses. (**d**) Comparison of paired pulse facilitation behavior for the different 2D heterostructure devices.

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
