# Peer review of "Artificial 2D van der Waals Synapse Devices via Interfacial Engineering for Neuromorphic Systems"

_nanomaterials, 2020, doi:10.3390/nano10010088_

Round 1
Reviewer 1 Report
The papers present a reliabe idea and proof of principle of synaptic behaviour of suitably designe conducting layers. It is interesting and worth to be published in present form in my opinion.
Author Response
The papers present a reliable idea and proof of principle of synaptic behavior of suitably designed conducting layers. It is interesting and worth to be published in present form in my opinion.
Response #1: We highly appreciated the reviewer’s positive comment.
Reviewer 2 Report
The paper is well written and the authors show interesting result.
However, some point need clarification and the later section of the pulsed voltage characterization requires some editing for clarity
What is the final thickness and layer number of the individual TMDs? The TEM/EDS figures appear to show the same magnification (the scale bars are different, however they appear to scale 1:1 – 5 x 2nm scaalebar = approx. 2 x 5nm scale bar). The Nb containing layer appear about 2nm apart, whereas the EDS map shows approx.. 5nm The Raman spectra shown in fig 2c) are apparently for the individual single layers. The question is if the distinct TMD peaks are preserved after transfer. Please also show the spectrum from a stacked structure as well. Please put the unit (nm) on the x-axis of figure 3b). What is the repeatability of the transfer double sweep curves? What is the sweep rate for the sweeps (V/s)? Is there any dependence on the number of sweeps or time in between sweeps? The very long discussion on page 5 line 163-173 is related to the contact to the WSe2 channel and not the hysteresis as formulated in the previous sentence. As it is out of context, discussion should be removed or put into the proper context (ohmic contact quality?). The schematic in figure 4 implies that holes transport (between S and D) takes place inside the NbO layer under negative bias. The authors may want to redraw the picture and also reformulate in the text that the trapped charge in the NbO is “depleting” the carriers in the WSe2 leading to lower current.Nevertheless, the proposed explanation regarding trapping in the NbO appears plausible.
Please write out EPSC when used the first time The reviewer is a novice in synaptic behavior. It is not clear what is seen in figure 5a) and 5d). It may be useful to indicate in the figures the regions with different gate voltages? Is t=0 the start of the first pulse in fig 5a? In fig. 5d) one would expect 5 regions, not 3 regions only? before the first pulse, during the pulse, between the pulses, during the 2nd pulse, after the pulse. What is the origin of the synaptic behavior without the NbO?Author Response
We would like to thank you, and the reviewers, for the time and effort spent in evaluating our manuscript and appreciate your favorable decision. The reviewers’ constructive comments have helped us improve our manuscript considerably. We have carefully addressed every issue raised by the referees; our responses to their individual comments are attached.
All revision parts in revised manuscript and supporting were marked with red color. Thus, we believe that these revisions certainly improve the quality of our manuscript much further. With these modifications, we would like to re-submit our manuscript to Journal of Nanomaterials.

Reviewer 3 Report
See attached file.

Author Response
We would like to thank you, and the reviewers, for the time and effort spent in evaluating our manuscript and appreciate your favorable decision. The reviewers’ constructive comments have helped us improve our manuscript considerably. We have carefully addressed every issue raised by the referees; our responses to their individual comments are attached. All revision parts in revised manuscript and supporting were marked with red color. Thus, we believe that these revisions certainly improve the quality of our manuscript much further. With these modifications, we would like to re-submit our manuscript to Journal of Nanomaterials.

Round 2
Reviewer 3 Report
In this revised manuscript, the Authors have substantially improved the quality and contents of their work. I am now convinced that the paper is suitable for publication in Nanomaterials.